# Causal Document-Grounded Dialogue Pre-training

**Yingxiu Zhao[1], Bowen Yu[2]\*, Haiyang Yu[2], Bowen Li[2], Jinyang Li[2],**
**Chao Wang[2], Fei Huang[2], Yongbin Li[2], Nevin L. Zhang[1]**

[1] The Hong Kong University of Science and Technology, [2] Alibaba Group

{yzhaocx,lzhang}@connect.ust.hk, libowen.ne@gmail.com,

{yubowen.ybw,yifei.yhy,ruiyi.wc,f.huang,shuide.lyb}@alibaba-inc.com

## Abstract

The goal of document-grounded dialogue (DocGD) is to generate a *response* by anchoring the *evidence* in a *supporting document* in accordance with the *dialogue context*. This entails four causally interconnected variables. While task-specific pre-training has significantly enhanced performances on numerous downstream tasks, existing DocGD methods still rely on general pre-trained language models without a specifically tailored pre-training approach that explicitly captures the causal relationships. To address this, we present the first causally-complete dataset construction strategy for developing million-scale DocGD pre-training corpora. Additionally, we propose a causally-perturbed pre-training strategy to better capture causality by introducing perturbations on the variables and optimizing the overall causal effect. Experiments conducted on three benchmark datasets demonstrate that our causal pre-training yields substantial and consistent improvements in fully-supervised, low-resource, few-shot, and zero-shot settings[1].

## 1 Introduction

Goal-oriented dialogue, focusing on assisting users in achieving their goals through natural language interactions, has made significant progress in recent years (Peng et al., 2021; He et al., 2022). Nonetheless, these systems frequently encounter constraints in providing information that extends beyond what can be obtained from particular databases or domains. To address this issue, researchers have proposed the goal-oriented Document-Grounded Dialogue (DocGD) task (Feng et al., 2020, 2021), which leverages documents as the external knowledge source to support dialogue systems in meeting the diverse information needs of users.

Recently, task-specific pre-training has shown an extraordinary ability to boost performances

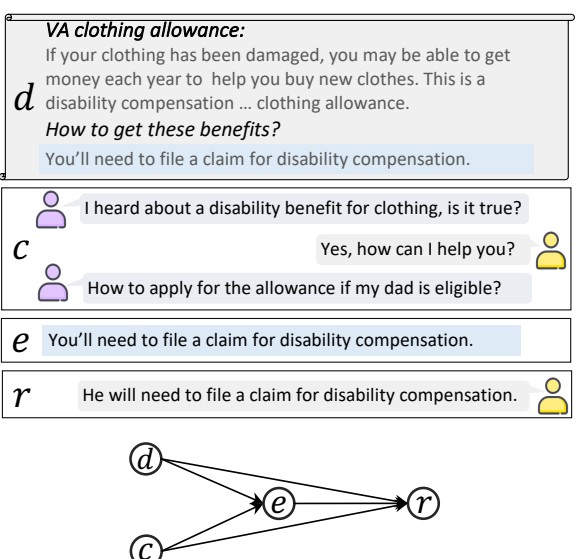

Figure 1: Causal graph of an example in DocGD, where four variables are causally connected: document $d$, evidence $e$, dialogue context $c$, and response $r$.

on downstream tasks, by mitigating the gaps between pre-training and fine-tuning due to different data distributions and training objectives (Mengge et al., 2020; Liu et al., 2021; Bai et al., 2022). Nonetheless, the study of pre-training for DocGD is hindered due to the difficulty of modeling the causal effects, which is a prominent characteristic of DocGD. As shown in Figure 1, the task requires the model to identify ***evidence*** in a ***document*** based on the dialogue ***context***, and then utilize the grounding evidence to generate a corresponding ***response***. This process involves the interplay of four variables that are causally connected. To attain precise modeling of causal effects during pre-training, two challenges must be overcome: (1) the scarcity of large-scale and causally-complete DocGD datasets for pre-training, as opposed to dialogue generation tasks that have the advantage of utilizing conversational data from various social media sources (Zhang et al., 2020), (2) the traditional likelihood objective (e.g., Raffel et al.

---

\* Corresponding author.

[1]The datasets and code will be made publicly available.

([2020](#)) being insufficient to capture the causal relationships among variables.

For the first challenge, we propose a novel strategy for building a DocGD pre-training corpus. We define a dataset as *causally-complete* if it includes all the variables related to a task and encompasses all reasonable causal relationships among these variables. Our strategy involves two steps. Firstly, we transform Wikipedia documents into dialogues by generating pseudo-user utterances and modifying evidence in the documents to serve as agent responses. Secondly, we extract grounding documents embedded in URLs and insert virtual evidence to supplement dialogues from Reddit. Both steps guarantee that the datasets are causally-complete, and they complement one another, as the former has authentic evidence with synthetic dialogues, while the latter possesses authentic dialogues with synthetic evidence.

To tackle the second challenge, we propose a causally-perturbed pre-training strategy that enhances the modeling of causality in our pre-training datasets. Our approach entails introducing causal interventions to both the document and evidence variables while optimizing the total effect of responses for different causes. The total effect comprises the natural direct effect (NDE) and total indirect effect (TIE) ([Niu et al., 2021](#)). In essence, the NDE quantifies the impact of irrelevant sentences in the supporting document, while the TIE captures the influence of evidence (detailed explanations in §3.3). Our objective is twofold: to enhance the model's resilience to perturbations in irrelevant sentences by minimizing the NDE, and to promote reliance on evidence in generating dialogue responses by maximizing the TIE. To achieve this, we retain relevant evidence while perturbing the remaining parts of the document to improve response consistency when using two versions, thus reducing the NDE. Additionally, we eliminate evidence from the document while preserving other information, subsequently decreasing the likelihood of generating original responses, thus maximizing the TIE.

Overall, we refer to the two aforementioned strategies jointly as Causal Document-Grounded Dialogue (CausalDD). We thoroughly conduct experiments and analyses on three DocGD benchmark datasets. Our results, obtained through fully-supervised, few-shot, low-resource, and zero-shot scenarios, and evaluated by both automatic and human assessment, convincingly demonstrate the

effectiveness of our pre-training corpus construction and causally-perturbed pre-training strategy. Especially, CausalDD even outperforms GPT-3.5.

## 2 Related Work

### 2.1 Causal Inference For NLP

Causal Inference is a statistical modeling tool that has been applied in explanatory analysis to better understand the relationships between variables ([Glymour et al., 2016](#); [Kuang et al., 2020](#); [Feder et al., 2022](#)). In the context of named entity recognition, [Zeng et al.2020](#) sought to eliminate spurious correlations between context and entity tokens by replacing entities with counterfactual tokens. [Wu et al.2020](#) similarly utilized counterfactual samples in sentiment classification, replacing causal terms with their antonyms. Our research endeavors to explore DocGD from a causal perspective, presenting the causal relationships among DocGD variables for the first time.

### 2.2 Document-Grounded Dialogue

Goal-oriented dialogue generation grounded in documents is a challenging and realistic task ([Ma et al., 2020](#); [Yu et al., 2022](#); [Zhang et al., 2023](#)). Researchers have increasingly utilized documents in a more flexible manner to improve the fluency and informativeness of model-generated responses, including in tasks such as Machine Reading Comprehension, Convention Question Answering, and the focus of this paper, DocGD. To support the development of models for these tasks, various datasets have been proposed, including CoQA ([Reddy et al., 2019](#)), QuAC ([Choi et al., 2018](#)), DoQA ([Campos et al., 2020](#)), Wizard ([Dinan et al., 2018](#)), Doc2dial ([Feng et al., 2020](#)), MultiDoc2Dial ([Feng et al., 2021](#)) and Doc2bot ([Fu et al., 2022](#)).

However, the high annotation requirements for document-grounded dialogues have limited the scale of available annotated data. To address this issue, [Li et al. (2020)](#) express the document knowledge as latent variables and devise a variational approach to achieve zero-resource knowledge-grounded dialogue generation. [Li et al. (2021)](#) homogenize different sources of knowledge (e.g., dictionaries, or knowledge graphs) into a unified representation to alleviate reliance on a single source. [Gao et al. (2022)](#) develop a prompt-connected multi-task learning to unify DocGD tasks. [Zhang et al. (2023)](#) propose coarse-to-fine knowledge selection to improve knowledge retrieval among

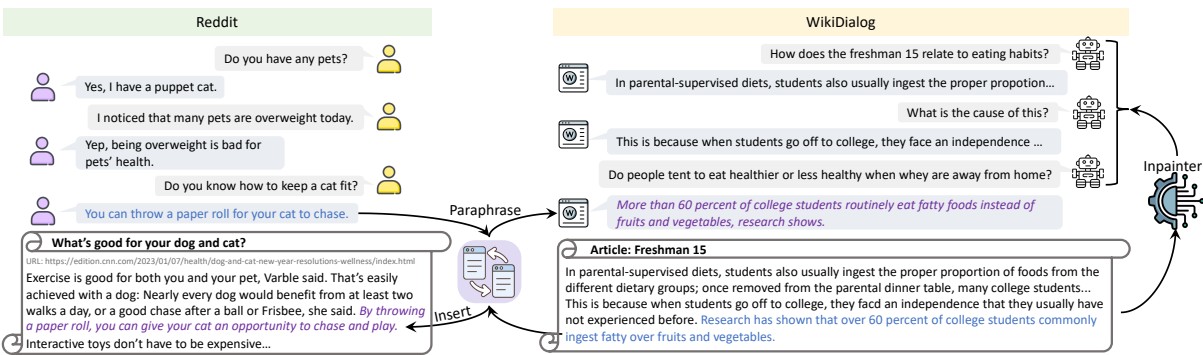

Figure 2: Two complementary datasets WikiDialog and Reddit built for DocGD, which are causally complete.

| Datasets | Dialogues | Documents | Total Turns |
|---|---|---|---|
| WikiDialog | 1.00M | 0.12M | 3.00M |
| Reddit | 1.00M | 1.00M | 1.39M |
| All | 2.00M | 1.12M | 4.39M |

Table 1: Statistics of CausalDD pre-training corpora.

multiple documents. These approaches omit pre-training on DocGD and merely initialize the parameters with general language models such as T5 (Raffel et al., 2020). Thus, how to effectively pre-train the DocGD model is still an open problem. In this paper, we give the answer from the perspective of causal inference and demonstrate that causal pre-training is effective in various DocGD settings.

## 3 Methodology

In this section, we first take a casual-effect perspective on the DocGD task in §3.1. We then propose two strategies to overcome the challenges discussed in §1: (1) a dataset construction strategy in §3.2 for building a causally-complete pre-training corpus; (2) a causally-perturbed pre-training strategy in §3.3 for better modeling causality of DocGD.

### 3.1 A Causal-Effect Perspective on DocGD

The DocGD task is commonly formulated as a sequential process comprising two sub-tasks: *knowledge grounding* and *response generation* (Feng et al., 2020; Gao et al., 2022). In knowledge grounding, a text segment denoted as $e$, is identified within the supporting document $d$, based on the dialogue context $c$. This segment serves as the evidence for generating the subsequent response $r$. Consequently, four variables $c, d, e, r$ are causally interrelated: The causal paths $d \rightarrow r$ and $c \rightarrow r$ directly influence the response, while an indirect effect occurs through the intermediary variable $e$. See Fig. 1 for an example causal graph.

### 3.2 Causally-complete Dataset Construction

While task-specific pre-training has been extensively researched in various domains and proven effective, there is a lack of research on pre-training for DocGD. The challenge lies in constructing a pre-training DocGD corpus that captures causal relationships among relevant variables. Merging a dialog corpus with unverified documents without careful consideration of causality can result in missing variables or weak causal connections. As a consequence, models may learn spurious features, such as generating responses solely based on the dialogue context $c$ while ignoring the document $d$. Our analysis (§4.7) demonstrates that this can significantly degrade performance.

To overcome this challenge, we propose a pre-training data construction strategy that utilizes high-quality documents from Wikipedia to create a causally-complete dataset with virtual dialogues. We further complement the corpus by leveraging real-world dialogues from Reddit to construct another dataset with virtual external knowledge. (Refer to data statistics in Table 1).

### 3.2.1 Causally-complete WikiDialog

Wikipedia offers a wealth of excellent articles, often authored or edited by experts who have invested considerable time and effort into ensuring clarity, accuracy, and addressing readers' queries. A distinctive feature of Wikipedia is that each page is dedicated to describing a specific entity. Therefore, when a user inquires about one entity from an agent, the corresponding page can be considered as the source of information for the agent's response.

We utilize this property to convert Wikipedia into two-person DocGD. Given a page, denoted as $d = (e_1, e_2, \ldots, e_m)$, consisting of $m$ sentences, each sentence is treated as evidence $e$, representing an agent's response in an $m$-round di-

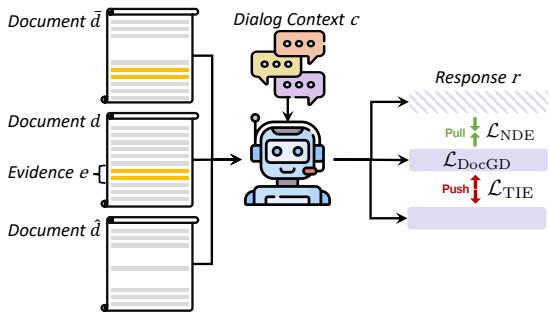

Figure 3: The diagram demonstrates causally-perturbed pre-training for CausalDD. The central document on the left represents the original document $d$, while the top and bottom documents represent perturbed documents $\bar{d}$ and $\hat{d}$ used for optimizing NDE and TIE, respectively.

alogue. To complete the dialogue, we employ a dialogue inpainter (Dai et al., 2022a) (see A.1) to generate missing user utterances, which are interleaved with the agent's responses. The inpainter generates a pseudo session $s_{\text{inpainter}} = (u_1, e_1, u_2, e_2, ..., u_m, e_m)$, where $u_i$ is the $i$-th generated utterance for the user, and $e_i$ is used as the agent's response. In our causally-complete WikiDialog, we first copy the first two turns from the inpainter. For the third turn, we randomly select one turn from the remaining turns as the utterance and grounding knowledge. To enhance naturalness, we employ a well-trained paraphrase model (see §A.2) to rewrite the evidence $e_3$ into the agent's response $r_3$, creating the dialogue sequence $s = (u_1, e_1, u_2, e_2, u_3, r_3)$. The dialogue context $c = (u_1, e_1, u_2, e_2, u_3)$, while the response $r_3$ is a paraphrased rendition of the evidence $e_3$.

**Remark.** Why is the above construction causally complete? First, the evidence $e$ is an exact sentence in the document $d$, and the dialogue context $c$ is generated by the dialogue inpainter based on $e$, so the evidence $e$ can be uniquely determined in $d$ by $c$, that is, $e$ is the effect of $\{c, d\}$. Considering response $r$ is a paraphrase of $e$, so $e$ is the direct cause of $r$, and the causal paths $c \rightarrow r$ and $d \rightarrow r$ can also be implicitly established.

### 3.2.2 Causally-complete Reddit

Despite having a causally-complete dataset like WikiDialog, the generated virtual dialogues may not fully align with the distribution of real-world human conversations. To address this issue, we propose supplementing the pre-training corpus with diverse and realistic conversations. We consider Reddit, a popular online platform for person-to-person discussions, as a valuable source of dialogue

context ($c$) and response ($r$), but lacking the document ($d$) and evidence ($e$) for DocGD. We observe that many submissions on Reddit contain URLs. These URLs often lead to web pages such as news articles that provide specific information related to the discussed topics. Therefore, we can crawl Reddit submissions that include URLs, using the content pointed to by the URL as the document ($d$), while utilizing the conversations and replies under the submission as $c$ and $r$ respectively.

**Remark.** Why is the above construction causally complete? First, the evidence $e$ is a paraphrase of response $r$, so a causal relationship can be established between $r$ and $e$. Given that $r$ is a natural response to the dialogue context $c$ on Reddit, causal paths $c \rightarrow r$ and $c \rightarrow e$ exist. Furthermore, through the random insertion of evidence $e$ into the document $d$, $d$ can thus become the cause of $e$, and then the cause path $d \rightarrow r$ could also be established.

### 3.3 Causal Pre-Training Framework

**DocGD-specific Pre-Training** To better capture the interdependence between evidence $e$ and response $r$, as described by Gao et al. (2022), we adopt the most efficient and straightforward fine-tuning method to sequentially generate the $e$ and $r$ based on the dialogue context $c$ and associated document $d$, instead of retrieving evidence and feeding it to the model to generate responses in a pipeline manner. We align our pre-training task with fine-tuning by optimizing the following objective:

$$\mathcal{L}_{\text{DocGD}} = - \sum_{(d,c,e,r) \in C} \log(p_\theta(e; r|d; c)) \quad (1)$$

where $C$ is the constructed causally-complete corpora in §3.2, $\theta$ is optimized during pre-training.

**Causally-perturbed Pre-Training** To facilitate the causal modeling of DocGD, we propose a causally-perturbed pre-training strategy by introducing causal perturbations to variables of DocGD and evaluating the outcomes under different causes.

Here, we utilize a common measurement, *causal effect*, to compare two potential outcomes for the same variable under two distinct treatment conditions (Rubin, 1978; Robins, 1986). Supposed that the random variable $X$ assigned with the observed value $x$, $X = x$, represents "under no-treatment condition" and $X = x^*$ represents "under treatment condition" (Niu et al., 2021). The *total effect* (TE) of treatment $X = x$ on the variable $Y$ compares two hypothetical situations $X = x$ and

$X = x^*$, which is denoted as: $\text{TE} = Y_{x^*} - Y_x$. In DocGD, we aim to estimate the effect of the document $d$ on the identification of evidence $e$ and the generation of response $r$. We denote $X = x$ to represent the original document $d$ (under no-treatment condition), and $X = x^*$ to represent applying perturbations to the document (under treatment condition). We use $Y$ to denote the generated sequence of evidence and response. More precisely, we further divide the document $d$ into two parts, the sentence $e$ where the evidence span lies and the other sentences $\{d \backslash e\}$ outside the evidence scope, i.e., $d = e \cup \{d \backslash e\}$. Hence, the total effect in DocGD can be written as:

$$\text{TE} = Y_{\{d \backslash e\}^*, e^*} - Y_{\{d \backslash e\}, e} \quad (2)$$

We adopt the decomposition in Niu et al. (2021) and adapt it to our causal DocGD scenario. Concretely, TE can be decomposed into the sum of the natural direct effect (NDE) and total indirect effect (TIE). NDE expresses the increase in the outcome $Y$ when $\{d \backslash e\}$ changes to $\{d \backslash e\}^*$:

$$\text{NDE} = Y_{\{d \backslash e\}^*, e} - Y_{\{d \backslash e\}, e} \quad (3)$$

TIE is the difference between TE and NDE:

$$\text{TIE} = Y_{\{d \backslash e\}^*, e^*} - Y_{\{d \backslash e\}^*, e} \quad (4)$$

For an ideal causal-aware DocGD pre-trained model, variations in $d \backslash e$ should not significantly affect the model's output, while the presence of evidence $e$ within $d$ determines the model's ability to identify evidence and generate responses that are relevant to the dialogue context. In other words, $Y_{\{d \backslash e\}^*, e}$ should be indistinguishable from $Y_{\{d \backslash e\}, e}$, while $Y_{\{d \backslash e\}^*, e^*}$ needs to differ significantly from $Y_{\{d \backslash e\}^*, e}$. Hence, it is necessary to improve the robustness of our model against perturbations on variables by minimizing NDE, and promote reliance on evidence in the generation of dialogue responses by maximizing TIE. (See the illustration of causally-perturbed pre-training of CausalDD in Figure 3.)

To achieve this, we design two causally-perturbed pre-training objectives. Firstly, we use Kullback-Leibler divergence to measure NDE:

$$\mathcal{L}_{\text{NDE}} = \sum_{(d,c,e,r) \in C} KL(p_\theta(e; r|d; c) || p_\theta(e; r|\overline{d}; c)) \quad (5)$$

where $\overline{d} = e \cup \{d \backslash e\}^*$ refers to disturbing the document $d$ by randomly deleting or inserting some sentences while keeping the evidence sentence $e$ in

$d$ retained. Secondly, to maximize TIE, we introduce the following unlikelihood loss:

$$\mathcal{L}_{\text{TIE}} = - \sum_{(d,c,e,r) \in C} \underbrace{\log(1 - p_\theta(e; r|\widehat{d}; c))}_{\text{unlikelihood}} \quad (6)$$

The situation $\widehat{d}$ here represents removing of evidence $e$ from the document $\overline{d}$, i.e., $\widehat{d} = \{d \backslash e\}^*$. After the removal, we aim to decrease the model's probability of generating tokens in the ground truth evidence $e$ and response $r$.

**Total Pretraining Objective** Overall, our pre-training objective is the sum of standard DocGD loss and our newly proposed causally-perturbed losses as follows:

$$\mathcal{L} = \mathcal{L}_{\text{DocGD}} + \mathcal{L}_{\text{NDE}} + \mathcal{L}_{\text{TIE}} \quad (7)$$

After pre-training, we fine-tune the obtained model on downstream datasets by optimizing $\mathcal{L}_{\text{DocGD}}$ in Eq. 1 following Gao et al. (2022).

## 4 Experiments

### 4.1 Datasets

We evaluate the effectiveness of our CausalDD for DocGD in both English and Chinese. For English, we use the two causally-complete datasets constructed in §3.2 for pre-training and evaluate the performances on two goal-oriented document-grounded dialogue datasets: Doc2dial (Feng et al., 2020) and MultiDoc2dial (Feng et al., 2021). Doc2dial (Feng et al., 2020) contains 3,474 dialogues with 44,149 turns for training and 661 dialogues with 8,539 turns for evaluation[2]. MultiDoc2dial (Feng et al., 2021) contains 4,796 dialogues with an average of 14 turns grounded in 488 documents, with multiple documents supporting each dialogue in four different domains.

For Chinese, we utilize a translation model (Wei et al., 2022) to translate the English pre-training data into Chinese, and evaluate the performance on a Chinese DocGD dataset Doc2bot (Fu et al., 2022). Doc2bot (Fu et al., 2022) has samples of Chinese conversations that are natural, coherent, and grounded in diverse documents. Although the translation model may impact the quality of Chinese pre-training data, we have observed significant improvements in our approach across three Chinese pre-trained backbones. We leave on constructing better Chinese pre-training data for future work.

---

[2]Since we cannot access the test set, we report results on the development set for comparison following previous work (Gao et al., 2022).

| Model | EM | F1 |
|---|---|---|
| BERTQA | 42.6 | 36.7 |
| BERT-PR-large | 44.2 | 38.9 |
| RoBERTa-PR-large | 55.7 | 54.6 |
| Multi-Sentence | 56.1 | 57.4 |
| DIALKI | 65.9 | 57.4 |
| UniGDD | 65.6 | 76.4 |
| GPT-3.5 | 46.1 | 57.3 |
| CausalDD | 66.0 | 77.3 |
| CausalDD$_{large}$ | **67.0** | **78.1** |

Table 2: Results on Doc2dial knowledge identification.

| Model | BLEU |
|---|---|
| DIALKI+BART-base | 25.8 |
| RoBERTa-PR-large+BART-base | 39.6 |
| RoBERTa-large+T5-base | 40.7 |
| UniGDD | 42.4 |
| GPT-3.5 | 3.57 |
| CausalDD | **43.0** |
| CausalDD$_{large}$ | 42.5 |

Table 3: Results on Doc2dial response generation.

## 4.2 Implementation Details

For pre-training, we use the pre-trained T5-base and more powerful T5-large (Raffel et al., 2020) to initialize CausalDD for English DocGD. For Chinese DocGD, we try three types of initialization: T5-Mengzi (Zhang et al., 2021), mT5 (Xue et al., 2021), and T5-Randeng (Wang et al., 2022). CausalDD is pre-trained on 4 80G NVIDIA A100 GPUs with a maximum learning rate of 1e-5 and a warm-up ratio of 0.1 for one epoch. The batch size per iteration is set to 8, and we use the AdamW optimizer with parameters beta1 = 0.9, beta2 = 0.98, and epsilon = 1e-6. The total pre-training time is about 60 hours. After CausalDD pre-training, the model is then fine-tuned on the Doc2dial, MultiDoc2dial, and Doc2bot datasets following Gao et al. (2022), with batch size of 4 and training epochs of 5. Note that both MultiDoc2dial and Doc2bot require the model to first retrieve relevant documents, identify evidence in those documents,

| Model | F1 | EM | BLEU |
|---|---|---|---|
| UniGDD$_{Randeng}$ | 38.6 | 39.6 | 15.3 |
| CausalDD$_{Randeng}$ | **42.9** | **41.8** | **22.3** |
| UniGDD$_{mT5}$ | 40.6 | 41.5 | 17.7 |
| CausalDD$_{mT5}$ | **48.2** | **47.4** | **22.0** |
| UniGDD$_{Mengzi}$ | 44.2 | 44.9 | 20.7 |
| CausalDD$_{Mengzi}$ | **48.5** | **47.8** | **25.4** |

Table 4: Results on Doc2bot.

and then generate corresponding responses. Hence, following Glass et al. (2022), we first train an additional retrieval model and a ranking model to locate documents relevant to dialogues as the external knowledge for downstream fine-tuning (see details in Appendix B.1).

## 4.3 Baselines

We compare CausalDD with several strong baselines, including UniGDD (Gao et al., 2022) and many commonly measured methods specific to each dataset. UniGDD utilizes the pre-trained T5 model (Raffel et al., 2020) as the initialization and optimizes $\mathcal{L}_{DocGD}$ in Eq. 1 on downstream datasets, which also serves as our most relevant baseline. Furthermore, we meticulously design the instruction to assess the performance of GPT-3.5. See more baselines and details in Appendix B.2.

## 4.4 Metrics

In concurrence with prevalent measurements (Gao et al., 2022; Feng et al., 2020, 2021), we utilize the metrics of Exact Match (EM) and token-level F1 for the identification of evidence and BLEU (Papineni et al., 2002) for the generation of responses.

## 4.5 Main Results

**Fully-Supervised** As shown in Tables 2, 3, 8, and 4, our proposed CausalDD method outperforms the baseline model on all evaluation metrics in both English and Chinese datasets. The improvements of CausalDD over other baselines are statistically significant with $p$-value $< 0.05$ under $t$-test. Regardless of the language or different T5 parameter initialization, CausalDD consistently surpasses the strongest baseline UniGDD.

Initialized with a larger T5, CausalDD$_{large}$ further enhances the performance compared to CausalDD, albeit with a slight decrease in BLEU score. To investigate this phenomenon, we compute the distinct scores with **Dist-n** following Li et al. (2015) to evaluate generated responses on the Doc2Dial dataset, shown in Table 7. We discover that the large model tends to generate more diverse responses, which may differ from the expressions of manually annotated answers. Furthermore, we observe that the popular large language model GPT-3.5 performs poorly on the existing DocGD datasets, despite including task instructions and cases in its prompt. Upon analyzing GPT-3.5's predictions, we find that its underperformance stems

| Dataset | Model | Few-Shot | | | Low-Resource | | |
|---|---|---|---|---|---|---|---|
| | | 5-Shot | 50-Shot | 100-Shot | 1% | 5% | 10% |
| **Doc2dial** | UniGDD | 13.7 | 14.8 | 17.1 | 17.8 | 36.9 | 39.8 |
| | CausalDD | 25.8 (12.1↑) | 27.4 (12.6↑) | 29.8 (12.7↑) | 34.7 (16.9↑) | 43.4 (6.5↑) | 46.9 (7.1↑) |
| **MultiDoc2dial** | UniGDD | 13.5 | 20.1 | 19.8 | 23.9 | 33.9 | 34.5 |
| | CausalDD | 29.8 (16.3↑) | 29.7 (9.6↑) | 30.6 (10.8↑) | 34.4 (10.5↑) | 42.0 (8.1↑) | 44.3 (9.8↑) |
| **Doc2bot** | UniGDD$_{Mengzi}$ | 0.00 | 0.01 | 0.10 | 0.00 | 2.3 | 12.6 |
| | CausalDD$_{Mengzi}$ | 12.3 (12.3↑) | 12.7 (12.7↑) | 14.9 (14.8↑) | 13.7 (13.7↑) | 24.8 (22.5↑) | 31.0 (18.4↑) |

Table 5: Few-Shot and Low-resource results for knowledge identification (F1 score).

| Dataset | Model | Few-Shot | | | Low-Resource | | |
|---|---|---|---|---|---|---|---|
| | | 5-Shot | 50-Shot | 100-Shot | 1% | 5% | 10% |
| **Doc2dial** | UniGDD | 2.00 | 2.27 | 3.07 | 5.07 | 14.6 | 15.6 |
| | CausalDD | 11.1 (9.1↑) | 11.2 (8.9↑) | 11.6 (8.6↑) | 14.9 (9.8↑) | 16.1 (1.5↑) | 18.7 (3.1↑) |
| **MultiDoc2dial** | UniGDD | 2.20 | 2.31 | 3.37 | 6.98 | 14.1 | 12.7 |
| | CausalDD | 14.0 (11.8↑) | 14.1 (11.8↑) | 13.9 (10.5↑) | 13.5 (6.5↑) | 14.7 (0.6↑) | 17.0 (4.3↑) |
| **Doc2bot** | UniGDD$_{Mengzi}$ | 0.00 | 0.00 | 0.00 | 0.00 | 2.10 | 2.31 |
| | CausalDD$_{Mengzi}$ | 2.50 (2.50↑) | 2.61 (2.61↑) | 3.70 (3.70↑) | 3.00 (3.00↑) | 12.3 (10.2↑) | 13.1 (10.8↑) |

Table 6: Few-Shot and Low-resource results for response generation (BLEU score).

| | Dist-1 | Dist-2 | Dist-3 | Dist-4 |
|---|---|---|---|---|
| UniGDD | 0.0736 | 0.3191 | 0.5055 | 0.6049 |
| CausalDD | 0.0736 | 0.3198 | 0.5079 | 0.6081 |
| CausalDD$_{large}$ | 0.0749 | 0.3308 | 0.5299 | 0.6347 |

Table 7: Distinct scores of responses on Doc2Dial

| Model | F1 | EM | BLEU |
|---|---|---|---|
| D$^{token}$-nq | 40.0 | 22.3 | 15.7 |
| D$^{struct}$-nq | 39.8 | 22.3 | 16.6 |
| D$^{token}$-ft | 43.6 | 26.4 | 18.8 |
| D$^{struct}$-ft | 43.5 | 26.1 | 19.5 |
| D$^{token}$-rr-cls-ft | 42.1 | 25.0 | 18.4 |
| D$^{struct}$-rr-cls-ft | 43.5 | 26.2 | 19.8 |
| CPII-NLP | 47.3 | - | 34.3 |
| R3 | 43.3 | - | 31.1 |
| G4 | 44.6 | - | 31.2 |
| Re3FiD | 46.7 | - | 33.5 |
| UniGDD | 61.5 | 45.8 | 31.8 |
| GPT-3.5 | 40.8 | 30.7 | 1.15 |
| CausalDD | 63.7 | 49.3 | **33.9** |
| CausalDD$_{large}$ | **64.5** | **51.0** | 33.6 |

Table 8: Results on MultiDoc2dial.

from a failure to adhere strictly to the given document's content for generating responses and providing evidence. Instead, it suffers from a severe hallucination issue, generating irrelevant content. We present the results of our human evaluation in Section 4.9.

We also observe that the model's improvements are more significant on Doc2bot in Table 4. We speculate that it's because the training data of Doc2bot is smaller than that of Doc2dial and MultiDoc2dial, and our model obtains better initialization for DocGD through causal pre-training, thus being more data-efficient for fine-tuning.

**Few-Shot & Low-Resource**    To verify the above speculation, we further conduct experiments on three datasets under few-shot and low-resource settings. We consider few-shot settings with only 5, 50, and 100 training examples and low-resource settings with 1%, 5%, and 10% of the original training datasets to train the models. The results are shown in Table 5, 13 and 6 for knowledge identification and response generation tasks, respectively (see more results in Appendix B.3). We can notice that the improvements of CausalDD are more significant in scenarios with such a small amount of training data. Specifically, an average improve-

ment of 9.7 points is achieved in these settings, indicating that our method is more effective with limited human-annotated data, which is particularly important for real-world applications.

**Zero-Shot**    We evaluate the performances on three datasets under the zero-shot setting. Results in Table 9 indicate that CausalDD achieves superior performances over UniGDD without any training samples of downstream tasks. This verifies the high quality of our constructed causally-complete corpora, and the good initialization provided by CausalDD for downstream fine-tuning.

### 4.6    Ablation Studies

To evaluate the contribution of each component in CausalDD, we conduct a series of ablation studies on two language datasets: Doc2dial and

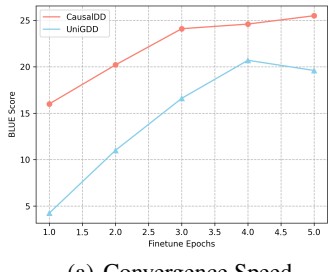
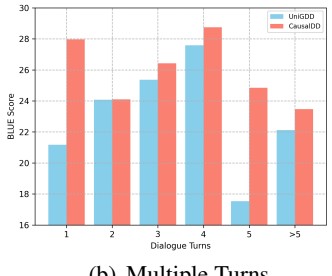
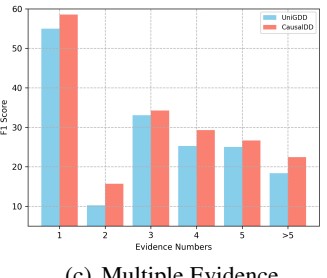

| (a) Convergence Speed | (b) Multiple Turns | (c) Multiple Evidence |
|---|---|---|

Figure 4: Compared to the UniGDD model, CausalDD has faster convergence time, better modeling of dialog history, and a stronger grounding of multiple evidence.

| Dataset | Model | F1 | EM | BLEU |
|---|---|---|---|---|
| Doc2dial | UniGDD | 13.9 | 0.00 | 1.98 |
| | CausalDD | 26.1 | 1.43 | 10.9 |
| MultiDoc2dial | UniGDD | 14.3 | 0.00 | 2.32 |
| | CausalDD | 30.0 | 2.76 | 13.9 |
| Doc2Bot | UniGDD$_{Mengzi}$ | 0.00 | 0.00 | 0.00 |
| | CausalDD$_{Mengzi}$ | 11.46 | 5.25 | 1.98 |

Table 9: Results under the zero-shot setting.

| Model | Doc2dial | | | Doc2bot | | |
|---|---|---|---|---|---|---|
| | F1 | EM | BLEU | F1 | EM | BLEU |
| Wikipedia | 76.9 | 65.7 | 42.5 | 46.8 | 46.5 | 23.6 |
| + Reddit | 77.2 | 65.8 | 42.7 | 47.6 | 47.0 | 24.8 |
| + NDE | 77.2 | 65.9 | 43.0 | 47.5 | 47.2 | 24.0 |
| + TIE | 77.0 | 66.0 | 42.8 | 46.8 | 46.6 | 23.9 |

Table 10: Ablation study of CausalDD on Doc2Dial.

Dot2Bot. The baseline for these studies is pre-training CausalDD exclusively on the causally-complete WikiDialog dataset. We then assess the impact of adding additional components, including (1) supplementing WikiDialog with our constructed Reddit dataset (+Reddit), (2) minimizing the NDE loss in Eq. 5 during pre-training (+NDE), and (3) maximizing the TIE loss in Eq. 6 (+TIE).

Results in Table 10 indicate that: (1) introducing a causally-complete Reddit containing real-world dialogues enhance the ability of the model to identify knowledge and generate better responses; (2) optimizing NDE to enhance the consistency of the model outputs with different support documents can enhance the robustness of the model; (3) optimizing TIE to prevent the normal output of the model when removing evidence from documents increases the model's reliance on the grounding evidence. These results validate that each component has a positive effect on CausalDD, leading to its better capability of modeling causal relationships among DocGD variables.

| Model | EM | F1 |
|---|---|---|
| UniGDD | 65.6 | 76.4 |
| CausalDD$_{incomplete}$ | 65.3 | 76.4 |
| CausalDD$_{complete}$ | 65.8 | 77.1 |

Table 11: Performance comparison with causal-incomplete and complete pre-training data on Doc2dial

To assess the effectiveness of our created complementary datasets, we also carry out a **case study** that compares the responses of CausalDD trained with various pre-training data (**Appendix B.4**).

### 4.7 Effects of Causally-complete Data

The creation of causally-complete pre-training data is one of the contributions of this paper. But (1) *is causally-complete data really necessary for DocGD pre-training?* (2) *what problems would arise if part of the causality in the pre-training data was missing?* To address these two questions, we build a causally-incomplete pre-training dataset by removing the introduced evidence $e$ from the previously-built Reddit dataset (i.e, the document $d = \{d \backslash e\}$). Then pre-training task is to generate responses $r$ based solely on documents and dialogue context $c$, without identifying knowledge first. We also pre-train a model using a causally-complete Reddit dataset for comparison. The results of Table 11 indicate that performance degrades when pre-training data cannot adequately model causal connections. The comparison with UniGDD (i.e., initialized with the original pre-trained T5) demonstrates that causally-incomplete pre-training introduces bias, resulting in a discrepancy between pre-training and fine-tuning.

### 4.8 Other Benefits of Causal Pre-training

In addition to overall performance improvement, we also observe some additional benefits brought by causal pre-training: (1) faster convergence speed

(Figure 4(a)), our model achieves good results in the first epoch due to having more DocDG-specific initialization parameters compared to general pre-training; (2) better modeling of dialog history (Figure 4(b)), we find better performance across all turns when we divided the Doc2bot test set based on the number of turns in the dialog history; (3) a better ability to ground complex evidence in Doc2bot, many samples require the model to ground multiple relevant segments in the document, and as the number of relevant evidence increases, CausalDD still shows better performance compared to UniGDD (Figure 4(c)).

### 4.9 Human Evaluation

To evaluate the performance of CausalDD against strong baselines, we randomly select 100 evaluation instances in Doc2Dial and request five human annotators to perform pairwise comparisons on two factors: (1) *Relevance*: indicating which response is more pertinent and relevant to the user's inquiry, and (2) *Informativeness*: determining which answer is more informative. See Table 12.

|  | Win | Tie | Lose |
|---|---|---|---|
| CausalDD vs. UniGDD | | | |
| Relevance | 42 | 55 | 3 |
| Informativeness | 43 | 47 | 10 |
| CausalDD vs. GPT-3.5 | | | |
| Relevance | 61 | 34 | 5 |
| Informativeness | 27 | 20 | 53 |

Table 12: Human Evaluation

Our method exhibits an apparent edge over UniGDD in two aspects when compared to baselines, highlighting the ability of CausalDD to effectively leverage rich document text to generate more suitable responses by capturing the causal relationship among variables. While GPT-3.5 can produce more informative responses, it depicts less consistency with the document and user's inquiry, implying the presence of hallucination issues.

### 5 Conclusion

In this paper, we demonstrate that modeling complete causal relationships among variables (i.e., documents, dialogue contexts, evidence, and responses) is necessary for pre-training for document-grounded dialogue task (DocGD). We propose a strategy for creating causally-complete pre-training datasets and design a causally-perturbed pre-training strategy to model the causality of

DocGD. To the best of our knowledge, this is the first work that analyzes DocGD from a causal perspective. Extensive experiments and analyses verify that our causal pre-training method CausalDD significantly improves performance under fully-supervised, few-shot, and low-resource settings, while also accelerating convergence and enhancing the ability to handle complex cases.

### 6 Limitations

Despite the fact that CausalDD has demonstrated its superior performance on three benchmarks, it still has a few limitations. Firstly, the pre-training data we construct is generated by models such as the dialogue inpainter and paraphrase model. Despite the large size of our causal-complete datasets, the data quality is slightly inferior to manually annotated data. We will also consider constructing data corpus through large language models like Li et al. (2023); Zhao et al. (2023). Secondly, there are other tasks such as knowledge graph-grounded dialogue, and our proposed pre-training data construction strategy may not be applicable. Lastly, the effectiveness of task-specific pre-training will decrease as the amount of labeled data increases, so if a large amount of DocGD labeled data is provided, the performance gains brought from our approach may be marginal.

### 7 Ethical Statement

This paper constructs a new pre-training dataset for DocGD, and we discuss some related ethical considerations here. First, in regards to intellectual property, the Wikipedia corpus and Reddit dump used as data sources are both freely available for research use, with the Wikipedia corpus shared under the CC BY-SA 3.0 license[3] and the Reddit dump shared for research purposes (Baumgartner et al., 2020). Second, we have taken measures to control potential risks by ensuring that the texts in Wikipedia do not contain private information. Additionally, we have ensured that the conversation data from Reddit does not include any personal information and that the topics discussed are public and harmless. Third, for human evaluation on the downstream Doc2Dial task, we hire five annotators to score 400 instances in total. The hourly pay is set to 15 US$ per person, higher than the local statutory minimum wage.

---

[3]https://creativecommons.org/licenses/by-sa/3.0

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

## A Method Details

### A.1 Dialogue Inpainter

The goal of a dialogue inpainting is to task a partial dialog to generate a complete dialog. A dialogue inpainter is trained using the following dialogue reconstruction task (Dai et al., 2022b): Given a complete dialog, $d = (u_1, u_2, \cdots, u_T)$, we randomly mask one utterance $u_t$, yielding a partial dialogue: $d_m(t) = (u_1, \cdots, u_{t-1}, \diamond, u_{t+1}, \cdots, u_T)$. With this partial dialogue as the input, we train T5 to predict $u_t$ with the following objective:

$$\mathcal{L} = -\sum_{d \in \mathcal{D}} E_{u_t \sim d}[\log p_\theta(u_t | d_m(t))], \quad (8)$$

where $\mathcal{D}$ is a corpus of complete dialogs and $u_t$ is a randomly sampled utterance from $d$. We then use the trained inpainter to transform a document into a dialog. Suppose a document $d = (s_1, s_2, \cdots, s_m)$, image each sentence $s_i$ is an utterance spoken by an agent in a dialogue with a user. We ask the inpainter to complete the following partial dialogue: $(\diamond, s_1, \diamond, s_2, \diamond, \cdots, \diamond, s_m)$. Each utterance from the imagined user starts masked and is responded to by the agent with a sentence from the document. We use the model autoregressively: generate $\hat{u_1}$ and replace the first mask $\diamond$, feed $(\hat{u_1}, s_1, \diamond, s_2)$ to complete the second mask. We continue the process until all masks are filled and the dialog is complete.

### A.2 Paraphrase Model

We adopt a well-trained paraphrase model from Alisetti (2020) to transform a sentence into another sentence with similar semantics. Specifically, the model takes an English sentence as input and produces a set of paraphrased sentences. We randomly select one sentence, and use it as the virtual utterance for causal-Wikidialogue and the virtual evidence for causal-Reddit, respectively.

## B Experiments Details

### B.1 Details of Retrieval and ranking

Because the Multidoc2dial and Doc2bot datasets do not give the document that the current dialog needs to be grounded but require the model to find the relevant document in the document corpus $\mathbb{Z}$, so we introduce an additional retrieval model and ranking model to find the most relevant document of the current dialogue context $c$ **Retrieve.** We use DPR (Karpukhin et al., 2020) as our retriever, which projects dialog context and documents to

a shared space using two BERT encoders (Kenton and Toutanova, 2019)). During retrieval, we perform a maximum inner-product search with FAISS (Lewis et al., 2020). Formally, we retrieve $K$ most relevant document $\mathbb{Z}_K = z_{[1,\cdots,K]} \in \mathbb{Z}$ for dialogue $c$ as:

$$\mathbb{Z}_K = \left\{ z_i \in \mathbb{Z} | \text{topK} \left\{ \text{BERT}(q)^\top \text{BERT}(z_i) \right\} \right\} \quad (9)$$

The goal of retrieval training is to develop an encoder that maps a given dialogue $d$ and all relevant documents into an embedding space such that the dialogue is close in proximity to its corresponding ground-truth document $z^+$. During training, we would like to maximize $P_{retr}(z^+|q, \mathbb{Z})$:

$$P_{retr}(z^+|q, \mathbb{Z}) = \frac{\exp(\text{sim}(q, z^+))}{\sum_{z \in \mathbb{Z}} \exp(\text{sim}(q, z))} \quad (10)$$

where $\text{sim}(q, z)$ is the cosine similarity between the normalized embeddings of the dialogue and document, generated by the BERT encoder. In order to perform contrastive learning, a set of negative documents must be sampled as it is not feasible to enumerate all other evidence. This is done by using the BM25 algorithm to retrieve the most difficult negative clue for each positive clue and then placing them into batches of 128 instances. The training loss is then calculated as the negative log-likelihood for the positive document.

**Rank.** The ranker we use is based on the sequence-pair classification. The dialogue $q$ and each candidate document $z_i \in \mathbb{Z}_K$ are input together to a BERT followed by a projection layer and Sigmoid function to calculate the ranking score of $z_i$:

$$s_i = \text{Sigmoid}(\text{Linear}(\text{BERT}(z_i \oplus q))\} \quad (11)$$

The training of the ranker begins by gathering the initial retrieval results on the training set. The top 36 samples (excluding the ground-truth evidence $z^+$) returned by the retrieval module are used as negative examples, and the ranker model is trained to distinguish positive cases from negative cases.

During inference, we first use the retrieval model to obtain the relevant document list and then use the rank model to identify the most relevant one as the supporting document. Note that we use the same retrieval and ranking model for CausalDD and our baseline UniGDD.

## B.2 Baselines

In Doc2dial, for the task of knowledge identification, we compare CausalDD with several strong baselines, including UniGDD (Gao et al., 2022), BERTQA (Kenton and Toutanova, 2019), BERT-PR (Daheim et al., 2021), RoBERTa-PR (Daheim et al., 2021), Multi-Sentence (Wu et al., 2021), and DIALKI (Wu et al., 2021). The other models formulate knowledge identification as a machine reading comprehension task and extract the grounding span from the document. For the response generation task, we compare CausalDD with UniGDD and several pipeline methods, including DIALKI+BART that uses DIALKI for knowledge identification and BART (Lewis et al., 2019) for response generation, RoBERTa-PR+BART, and RoBERTa+T5 (Gao et al., 2022).

In MultiDoc2dial, we first use the same Retrieval and Ranking module as Re2G (Glass et al., 2022) to obtain relevant documents as input for UniGDD and CausalDD. We also compare a series of baselines set up by Feng et al. (2021), which use BM25 and multiple DPR variances as retrievers, and use a BART-large pre-trained on the CNN dataset as the generation module. Moreover, we compare our method CausalDD with recent methods: R3 (Bansal et al., 2022), G4 (Zhang et al., 2022), and CPII-NLP (Li et al., 2022) proposed in the DialDoc Workshop and Re3FiD (Zhang et al., 2023), considering these three methods did not provide the exact-match results, we leave blanks in the Table 8.

In Doc2bot, we mainly compare with UniGDD and use three different T5 pre-trained models T5-Mengzi (Zhang et al., 2021), mT5 (Xue et al., 2021), and T5-Randeng (Wang et al., 2022). to initialize for a more comprehensive comparison.

The prompt for GPT-3.5 is carefully designed to match the input-output format of the training dataset of Doc2Dial. Examples and test input will be filled in the brackets as the instruction.

```
Document-grounded dialogue task aims
to identify evidence from a supporting
document for a dialogue between a user
and an agent for answering the user's
question.      Then   the   agent   replies
to  the  user  based  on  the  retrieved
evidence. Here are two examples for your
reference.  In  the  example-input,  <last-
turn> refers to the last utterance of the
user.  After <last-turn> part, a reverse
order of a dialogue between <user> and
```

```
<agent> is provided.  After the signal
</title>,  the  supporting  document  is
provided. Specifically, you need to first
retrieve the evidence (i.e.<grounding>)
from  the  document  in  the  test-input
based  on  the  dialogue  and  the  user's
query. Sentences after <grounding> must
be  the  exact  same  string  in  the  document
(including  the  spaces  and  punctuation).
Then  you  should  continually  generate
the  response  (i.e.<agent>)  based  on  the
evidence  as  an  agent  to  reply  to  the
user.  Example-1: {example1} Example-2:
{example2}   Test-Input:    {test-input}
Test-Output:
```

### B.3 More Results of Few-Shot & Low Resource

We present experimental results of CausalDD and the strongest baseline UniGDD in Table 13 and 6.

### B.4 Case Study

To assess the effectiveness of our created complementary datasets, we compare the responses of CausalDD under various data scenarios: after pre-training only on WikiDialog, only on Reddit, and on their combined corpora followed by Doc2dial fine-tuning.

From the case in Figure 5, we can refer that:

- UniGDD is able to accurately identify the grounding evidence in the supporting document, however, the generated response is just a simple copy of the evidence.

- After training solely on WikiDialog, the predicted response is more fluent and more consistent with the dialogue context rather than the copy of the evidence. This verifies the high quality of our constructed causally complete WikiDialog.

- After training solely on Reddit, the response is more colloquial while retaining high quality.

- Pre-training the complementary datasets (i.e.,WikiDialog + Reddit) with CausalDD, the generated response is more precise and natural compared with the ground truth. This demonstrates that constructing complementary datasets that are both causally complete yields better performances for downstream tasks' fine-tuning.

| Dataset | Model | Few-Shot | | | Low-Resource | | |
|---------|-------|----------|--|--|--------------|--|--|
| | | 5-Shot | 50-Shot | 100-Shot | 1% | 5% | 10% |
| **Doc2dial** | UniGDD | 0.00 | 0.00 | 0.00 | 0.38 | 6.11 | 3.43 |
| | CausalDD | 1.50 (1.50↑) | 1.95 (1.95↑) | 2.49 (2.49↑) | 6.16 (5.78↑) | 15.4 (9.29↑) | 20.1 (16.7↑) |
| **MultiDoc2dial** | UniGDD | 0.00 | 0.24 | 0.07 | 0.00 | 9.42 | 3.95 |
| | CausalDD | 2.93 (2.93↑) | 3.02 (2.78↑) | 3.31 (3.24↑) | 7.24 (7.24↑) | 14.5 (5.08↑) | 17.1 (13.2↑) |
| **Doc2bot** | UniGDD$_{Mengzi}$ | 0.00 | 0.00 | 0.00 | 0.00 | 1.18 | 11.7 |
| | CausalDD$_{Mengzi}$ | 5.83 (5.83↑) | 5.99 (5.99↑) | 6.36 (6.36↑) | 5.52 (5.52↑) | 21.0 (19.8↑) | 28.7 (17.0↑) |

Table 13: Few-Shot and Low-resource results for knowledge identification (EM score).

### Dialogue Context

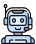 I filled out all of the information in the Retirement Estimator and it took a long time. When I came back from answering the door, all of the information was gone. What happened?

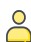 Oh that's too bad. Were you gone for a long time?

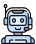 Yes I guess I was.

### Supporting Document

...... How Long Can You Stay On Each Page? *For security reasons, there are time limits for viewing each page.* *You will receive a warning after 25 minutes without doing anything, and you will be able to extend your time on the page.* After the third warning on a page, you must move to another page. If you do not, your time will run out and your work on that page will be lost.

### Response:

*UniGDD:*

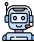 For security reasons, there are time limits for viewing each page. You will receive a warning after 25 minutes without doing anything and you will be able to extend your time on the page.

*WikiDialog:*

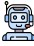 Then you should know that there are time limits for viewing each page. You will receive a warning after 25 minutes without doing anything and you will be able to extend your time on the page.

*Reddit:*

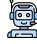 Okay, for security reasons there are time limits for viewing each page. You will receive a warning after 25 minutes without doing anything and you will be able to extend your time on the page.

*WikiDialog+Reddit (CausalDD):*

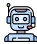 Do you know that for security reasons there are time limits for viewing pages?

*Ground Truth:*

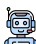 For reasons of security, there are time limits for viewing each page.

Figure 5: Case study for Doc2dial.