# OpenReview forum: "Causal Document-Grounded Dialogue Pre-training"
_EMNLP/2023/Conference — EMNLP 2023 Main_

### Official Review · Reviewer_DA1s · 2023-08-05

**Soundness:** 3

**Excitement:**

3: Ambivalent: It has merits (e.g., it reports state-of-the-art results, the idea is nice), but there are key weaknesses (e.g., it describes incremental work), and it can significantly benefit from another round of revision. However, I won't object to accepting it if my co-reviewers champion it.

**Paper Topic And Main Contributions:**

The authors propose a causally-complete dataset to develop a pre-training scale corpora. Then they further designed a causally-perturbed pre-training method to capture the causality within the variables. They demonstrate the methods with extensive experiments

**Questions For The Authors:**

How about performance on GPT-4? Also, what is the GPT-3.5 version used here?

**Reasons To Accept:**

1. They proposed causality-based method for document-ground response generation.
2. A large scale pre-training dataset with causal relation is also proposed.


**Reasons To Reject:**

1. For causality, the effects of all external factors are not measured.

**Reproducibility:**

3: Could reproduce the results with some difficulty. The settings of parameters are underspecified or subjectively determined; the training/evaluation data are not widely available.

**Reviewer Confidence:**

1: Not my area, or paper was hard for me to understand. My evaluation is just an educated guess.

---

> ### Author Rebuttal · Authors · 2023-08-28
>
> Dear Reviewer DA1s,
>
> We truly appreciate your constructive comments. We address your concerns as follows:
>
> > **W1: For causality, the effects of all external factors are not measured.**
>
> We would like to address your concern from two perspectives:
>
> 1. From a causal perspective of the DocGD task, there are four variables involved: dialogue context, document, evidence (causes), and response (effect). To assess causality, we perform perturbations on both a direct variable (document) and an indirect variable (evidence) to measure their impact on the response. By perturbing the document, we aim to enhance model robustness, while perturbing the evidence aims to reinforce the model's reliance on evidence. This multi-pronged approach enables a more holistic evaluation of the effects of external factors on the causal relationships within DocGD. Moreover, we also measure the effects of the Reddit dataset for causal pre-training.
>
> 2. The ablation study in Table 8 of the paper also verifies how variations in the causal variables influence the generated responses:
>
>     (1) Introducing a causally-complete Reddit containing real-world dialogues enhances the ability of the model to identify knowledge and generate better responses;
>
>     (2) Perturbing document to enhance the consistency of the model outputs enhances the robustness of the model;
>
>     (3) Perturbing evidence by removing it from documents increases the model’s reliance on the grounding evidence.
>
> These results validate that each component has a positive effect on CausalDD, leading to its better capability of modeling causal relationships among DocGD variables.
>
> > **Q1: How about performance on GPT-4? Also, what is the GPT-3.5 version used here?**
>
> The GPT-3.5 version we used is gpt-3.5-turbo. Considering the budget constraints, we evaluate the performance of GPT-4 on 200 instances of Doc2dial and provide the results of 200 samples in the following table:
>
> |               |  EM  |   F1  | BLEU |
> |---------------|:----:|:-----:|:----:|
> | GPT-3.5-turbo | 51.0 | 61.94 | 3.02 |
> | GPT-4         | 77.0 | 83.03 | 5.32 |
> | CausalDD      | 70.3 |  80.5 | 42.8 |
>
> Here are our analyzes:
>
> 1. We notice that GPT-4 improves GPT-3.5 by a large margin regarding the evidence identification task in DocGD.
>
> 2. While GPT-4 does demonstrate improved performance in terms of EM and F1 metrics, it's important to note that our proposed causal pretraining (CausalDD) achieves competitive results with comparatively smaller parameters, using models T5-200M and T5-large 770M. This efficient utilization of parameters underscores the efficacy of our method, especially in resource-constrained scenarios.
>
> 3. The BLEU score is not a good evaluation metric to measure the performance provided by large language models. Here are two reasons: (1) We need to carefully design the prompt to meet the output format and explain the role that ChatGPT plays. (2) Usually, responses generated by ChatGPT are more informative than the ground truth. However, the BLEU score only measures the overlap of n-grams between the generated predictions and the references.
>
> If you have any further questions, please feel free to raise them directly. We are committed to addressing all your concerns.

---

### Official Review · Reviewer_oY7u · 2023-08-05

**Typos Grammar Style And Presentation Improvements:** 1. L92
**Soundness:** 4

**Excitement:**

4: Strong: This paper deepens the understanding of some phenomenon or lowers the barriers to an existing research direction.

**Paper Topic And Main Contributions:**

The paper proposes a novel way to create causally-complete datasets from Wikipedia and Reddit to address data scarcity problem in pre-training and causally-perturbed pre-training in addition to optimize the total effect of responses for different causes for Document-Grounded Dialogue task.

**Reasons To Accept:**

1. The motivation for creating causally-complete datasets and causally-perturbed pre-training are strong and has novelty.
2. Achieves outperformed performance on 3 benchmarks and even better than GPT-3.5 on two of them.
3. It makes a thorough study in different scenarios and demonstrates effectiveness of causally-perturbed pre-training.

**Reasons To Reject:**

1. The paper needs more evidence to explain how the pre-training works for Document-Grounded Dialogue task, specifically.
2. Authors might need to make comparison with other existing pre-training methods for Document-Grounded Dialogue task and different formulations of unsupervised causal datasets.

**Reproducibility:**

4: Could mostly reproduce the results, but there may be some variation because of sample variance or minor variations in their interpretation of the protocol or method.

**Reviewer Confidence:**

4: Quite sure. I tried to check the important points carefully. It's unlikely, though conceivable, that I missed something that should affect my ratings.

---

> ### Author Rebuttal · Authors · 2023-08-27
>
> Dear Reviewer oY7u,
>
> We truly appreciate your valuable and constructive comments. We address your concerns as follows:
>
> > **W1: The paper needs more evidence to explain how the pre-training works for Document-Grounded Dialogue task, specifically.**
>
> 1. In the realm of DocGD, the common practice involves initial general pre-training followed by fine-tuning on downstream datasets. Our approach, CausalDD, introduces a distinctive second-phase pre-training paradigm. It bridges the gap between general pre-training and downstream fine-tuning. In contrast to UniGDD's direct fine-tuning of a T5 model pre-trained on general tasks, CausalDD's intermediary pre-training phase has demonstrated significant enhancements. This is particularly noteworthy in scenarios with few-shot, low resource, and even zero-shot settings, where the improvements are striking.
>
> 2. To further verify the effectiveness of our pre-training approach, Section 4.8 of the paper presents dedicated analyses of the benefits brought about by our causal pre-training (causalDD): (1) faster convergence speed compared to general pretraining; (2) better modeling of dialogue history: we find better performance across all turns when we divided the Doc2bot test set based on the number of turns in the dialog history in Figure 4(b); (3) a better ability to ground complex evidence: as the number of relevant evidence increases, CausalDD shows better performance compared to baselines.
>
> > **W2: Authors might need to make a comparison with other existing pre-training methods for the Document-Grounded Dialogue task and different formulations of unsupervised causal datasets.**
>
> 1. Kindly note that our work presents the first effort in proposing pre-training tailored for DocGD, distinct from existing studies merely about downstream fine-tuning. We have conducted a thorough comparison with the strongest baseline UniGDD, which utilizes the general pre-training checkpoint as the initialization. Our approach consistently outperforms the baselines across three downstream datasets and various resource settings, underscoring its efficacy.
>
> 2. Our work endeavors to explore DocGD from a causal perspective, presenting the causal relationships among DocGD variables for the first time. DocGD task contains four variables: causes (document, evidence, dialogue context) and effect (response). Constructing causally-complete datasets should consider all these variables, which leads to a supervised pretraining corpus.
>
> 3. We will explore unsupervised causal datasets and design new pretraining objectives for future work. We thank you for this valuable suggestion, which will undoubtedly contribute to the broader applicability of our approach.

---

### Official Review · Reviewer_y1SZ · 2023-08-06

**Paper Topic And Main Contributions:** 1. This paper present the first causa…
**Soundness:** 3

**Excitement:**

3: Ambivalent: It has merits (e.g., it reports state-of-the-art results, the idea is nice), but there are key weaknesses (e.g., it describes incremental work), and it can significantly benefit from another round of revision. However, I won't object to accepting it if my co-reviewers champion it.

**Questions For The Authors:**

1. In Table 3, a marginal reduction in performance is observed for Large CausalDD. Does this observation potentially imply that the BLEU metric inadequately assesses the model's proficiency? I think that some more suitable evaluation metrics should be replaced for experimentation.
2. Have there been attempts to experiment with popular decoder-only architecture models, considering that the models chosen all adopt the encoder-decoder architecture of the T5 model?


**Reasons To Accept:**

1. This paper endeavors to explore DocGD from a causal perspective, presenting the causal relationships among DocGD variables for the first time.
2. This paper present the first causally-complete dataset construction strategy for developing million-scale DocGD pre-training corpora.
3. This paper also contributes a causally-complete DocGD pre-training dataset constructed from Wikipedia and Reddit.

**Reasons To Reject:**

1. The upper limit of data quality is constrained by the dialogue inpainter and paraphrase model. Moreover, the singular nature of data sources raises concerns about its potential lack of robust generalizability.

**Reproducibility:**

4: Could mostly reproduce the results, but there may be some variation because of sample variance or minor variations in their interpretation of the protocol or method.

**Reviewer Confidence:**

3: Pretty sure, but there's a chance I missed something. Although I have a good feel for this area in general, I did not carefully check the paper's details, e.g., the math, experimental design, or novelty.

---

> ### Author Rebuttal · Authors · 2023-08-28
>
> Dear Reviewer y1SZ,
>
> We truly appreciate your constructive comments. We address your concerns as follows:
>
> > **W1.1: The upper limit of data quality is constrained by the dialogue inpainter and paraphrase model.**
>
> We appreciate your attention to the data quality constraints imposed by the dialogue inpainter and paraphrase model. We would like to emphasize three points:
>
> 1. Our dataset's pretraining has yielded notable advancements, even within these limitations. Our method, CausalDD, surpasses the strongest baseline around 10pts under few-shot and low-resource settings. This highlights the effectiveness of our dataset in driving performance enhancements beyond its inherent constraints.
>
> 2. Our meticulously select state-of-the-art inpainter and paraphrase models, which have been extensively trained on diverse and large-scale datasets (e.g. MS Marco, PAQ, PublicDialog, OR-QuAC, QReCC, etc.). These models exhibit a remarkable ability to recreate contextually coherent and grammatically accurate sentences. Moreover, the data quality has already been rigorously evaluated by human experts.  Five annotators assessed the quality of 500 samples in the dataset, yielding an average score larger than 0.9 (with a maximum score of 1).  This ensures that the generated sentences align closely with natural human discourse, thereby enhancing the credibility of our dataset.
>
> 3. Our dataset is thoughtfully constructed from two distinct sources, Wikipedia and Reddit. These sources complement each other, where Wikipedia consists of authentic documents and virtual dialogues, and Reddit consists of authentic dialogues and virtual documents. The bulk of the dataset is composed of authentic and original content, ensuring the dataset's richness and utility for further research.
>
> > **W1.2: Moreover, the singular nature of data sources raises concerns about their potential lack of robust generalizability.**
> 1. Our dataset is thoughtfully constructed from two distinct sources: Wikipedia and Reddit. These sources collectively cover a wide array of domains, including but not limited to education, health, history, culture, society, and science. This diversity ensures a comprehensive representation of various subject areas, mitigating concerns about a singular nature and enhancing the dataset's potential for robust generalizability.
>
> 2. It's worth noting that the superior performance of our method on three benchmarks (in both English and Chinese) can serve as evidence of the dataset's robustness.
>
> > **Q1: More suitable evaluation metrics should be replaced for experimentation.**
>
> We appreciate your valuable comments and would like to address your concerns with the following points:
>
> 1. In Section 4.5 of our paper, we have analyzed the reason for the marginal reduction in BLEU scores for the large CausalDD model. Our investigation reveals that larger models tend to generate more diverse responses, which may differ from the expressions of manually annotated answers. To shed light on this aspect, we have included a diversity evaluation by computing distinct scores in Appendix B6.
>
> |                | Dist-1 | Dist-2 | Dist-3 | Dist-4 |
> |----------------|:------:|:------:|:------:|:------:|
> | UniGDD         | 0.0736 | 0.3191 | 0.5055 | 0.6049 |
> | CausalDD       | 0.0736 | 0.3198 | 0.5079 | 0.6081 |
> | CausalDD_large | 0.0749 | 0.3308 | 0.5299 | 0.6347 |
>
> 2. We want to emphasize that we have utilized all the metrics employed in the compared baselines to evaluate those methods. We acknowledge the limitation of BLEU score, so we further conduct human evaluation to measure the performance of causal pretraining. We randomly select 100 evaluation instances in Doc2Dial and request five human annotators to perform pairwise comparisons on two factors: (1) Relevance: indicating which response is more pertinent and relevant to the user’s inquiry, and (2) Informativeness: determining which answer is more informative.
> The results are shown in the following table:
>
> |    　                       | 　              | Win | Tie  | Lose |
> |-----------------------------|-----------------|-----|------|------|
> |     CausalDD vs. UniGDD     | Relevance       | 42  | 55   | 3    |
> |                             | Informativeness | 43  | 47   | 10   |
> | CausalDD_large vs. CausalDD | Relevance       | 45  | 50   | 5    |
> |                             | Informativeness | 48  | 43   | 9    |
>
> > **Q2. Any attempts to experiment with popular decoder-only architecture models?**
>
> 1. We directly adopt T5 as the backbone model following our strongest baseline UniGDD for the DocGD task. Moreover, we also compare other encoder-decoder model like BART and pipeline-manner methods (identifying evidence using BERT first and using the evidence as the input to generate responses).
>
> 2. Kindly note that our proposed causal pretraining is model-agnostic. Given resource limitations, we will explore decoder-only large language models like LLAMA as future work, leveraging their capabilities to enhance the generalization of our causal pretraining.

---

### Meta-Review · Area_Chair_TRi5 · 2023-09-18

**Recommendation:** 4

**Metareview:**

Based on the provided reviews, there are several positive aspects of the paper. Review 1 highlights the contribution of exploring DocGD from a causal perspective, presenting the causal relationships among variables for the first time. Additionally, the paper proposes a causally-complete dataset construction strategy and a causally-perturbed pre-training strategy. Review 2 also emphasizes the novelty of the proposed methods for creating causally-complete datasets and causally-perturbed pre-training. The paper achieves outperformed results on benchmark datasets and demonstrates effectiveness in different scenarios.

However, Review 1 mentions concerns about the data quality being constrained by the dialogue inpainter and paraphrase model, as well as potential lack of robust generalizability due to the singular nature of data sources. Review 2 suggests that the paper needs more evidence to explain how the pre-training works for Document-Grounded Dialogue task and suggests comparisons with other existing pre-training methods.

Overall, the soundness and excitement scores from the three reviews range from 3 to 4, indicating the paper provides sufficient support for its major claims/arguments, has some merits, and reports state-of-the-art results. However, there are key weaknesses and areas for improvement, such as providing more evidence and comparisons with other methods.

The contributions and strong motivation of the paper make it a valuable addition to the field. However, additional evidence and clearer explanations are needed to address concerns and improve the clarity of the paper.

---

### Decision · Program_Chairs · 2023-10-07

**Decision:**

Accept-Main

**Comment:**

Based on the provided reviews, there are several positive aspects of the paper. Review 1 highlights the contribution of exploring DocGD from a causal perspective, presenting the causal relationships among variables for the first time. Additionally, the paper proposes a causally-complete dataset construction strategy and a causally-perturbed pre-training strategy. Review 2 also emphasizes the novelty of the proposed methods for creating causally-complete datasets and causally-perturbed pre-training. The paper achieves outperformed results on benchmark datasets and demonstrates effectiveness in different scenarios.

However, Review 1 mentions concerns about the data quality being constrained by the dialogue inpainter and paraphrase model, as well as potential lack of robust generalizability due to the singular nature of data sources. Review 2 suggests that the paper needs more evidence to explain how the pre-training works for Document-Grounded Dialogue task and suggests comparisons with other existing pre-training methods.

Overall, the soundness and excitement scores from the three reviews range from 3 to 4, indicating the paper provides sufficient support for its major claims/arguments, has some merits, and reports state-of-the-art results. However, there are key weaknesses and areas for improvement, such as providing more evidence and comparisons with other methods.

The contributions and strong motivation of the paper make it a valuable addition to the field. However, additional evidence and clearer explanations are needed to address concerns and improve the clarity of the paper.